# First Record of *Osphya* (Melandryidae: Osphyinae) from Chinese Mainland Based on Morphological Evidence and Mitochondrial Genome-Based Phylogeny of Tenebrionoidea †

Haoyu Liu [1], Lilan Yuan [1,2], Ping Wang [2], Zhao Pan [1], Junbo Tong [1], Gang Wu [3,*] and Yuxia Yang [1,*]

1    Key Laboratory of Zoological Systematics and Application, School of Life Science, Institute of Life Science and Green Development, Hebei University, Baoding 071002, China
2    College of Agriculture, Yangtze University, Jingzhou 434025, China
3    College of Plant Science and Technology, Huazhong Agricultural University, Wuhan 430070, China
*    Correspondence: wugang@mail.hzau.edu.cn (G.W.); yxyang@hbu.edu.cn (Y.Y.)
†    This published work and the nomenclatural acts it contains have been registered in ZooBank, the Online Registration System for the ICZN (International Code of Zoological Nomenclature). The LSID (Life Science Identifier) for This Publication is: LSID urn:lsid:zoobank.org:pub:5F023DB4-BA5A-40A7-99F6-8145BA71B6A7.

**Abstract:** *Osphya* Illiger (Melandryidae: Osphyinae) as a species-poor insect group, exhibits a widespread distribution in the Northern Hemisphere, however, the research of the genus is poorly documented especially in East Asia. Herein, an interesting species is discovered in Shennongjia National Natural Reserve (Hubei, China). The examination of morphological characters and comparisons with others show it to be a new member of *Osphya*, which is described under the name of *O. sinensis* sp. n. The characteristic photos and a key to the species of *Osphya* from East Asia are provided. Meanwhile, the mitochondrial genome of *O. sinensis* sp. n. is sequenced and annotated. Based on this obtained mitogenome and the publicly available data, we reconstructed the phylogeny of Tenebrionoidea by different cladistics methods to investigate the relationships between the new species with others. The results consistently recover *O. sinensis* sp. n. sister to *O. bipunctata* (Fabricius) with high supporting values, which further confirm the placement of the new species in the genus *Osphya*. This is the first time reporting the genus *Osphya*, the only representative genus of melandryid Osphyinae from mainland China, which enriches the diversity of beetles from the Chinese fauna at both generic and subfamilial levels.

**Keywords:** taxonomy; identification key; distribution map; beetles diversity; Chinese fauna





## 1. Introduction

The genus *Osphya* Illiger, 1807 belongs to the subfamily Osphyinae of the family Melandryidae in the superfamily Tenebrionoidea [1,2]. To date, it contains 27 species, which are strictly confined in the Northern Hemisphere and widely distributed in Palaearctic (17 species), Nearctic (3 species), Neotropical (3 species), and Oriental regions (4 species) [3,4]. The taxonomy of *Osphya* has been neglected almost for a century, except for a few species from Europe reported in recent years [3,4]. Most species remain poorly known at present, and except for the simple original descriptions [5–17], no subsequent data were added even for the distribution records. Although the Chinese mainland has a large area and is recognized as one of the global biodiversity hotspots [18], no *Osphya* species were recorded in this area except for two species from Taiwan Island (*O. trilineata* Pic, 1910 and *O. formosana* Pic, 1927). The absence of *Osphya* species in the Chinese mainland is bewildering despite the hard work endeavored by Chinese entomologists for many years. Herein, interesting material from Shennongjia National Natural Reserve, Hubei Province, China, was obtained. Its morphological characters seemed to match the diagnosis of *Osphya*. The genus is characterized by the abruptly narrowed head, latero-basal impressions on pronotum,

tarsal formula 5-5-4, bidentate tarsal claws, penultimate tarsomeres with distinct lobes, extended under last tarsomeres, and elongated fist hind tarsomeres, as well as the structure of the aedeagus [3,4,19,20]. Further comparison with others (*O. orientalis* (Lewis, 1895), *O. albofasciata* Champion, 1916, *O. melina* Champion, 1916, *O. nilgrica* Champion, 1916, *O. nigriventris* Champion, 1920, *O. nigroapicalis* Pic, 1921, *O. dissimilis* Champion, 1922, *O. harmandi* Pic, 1926, *O. rufa* Pic, 1927, and *O. superba* Pic, 1927) from the adjacent areas showed it as an unknown species of *Osphya*, which will be described herein.

Meanwhile, we successfully sequenced and annotated the mitochondrial genome of the new species. The mitochondrial genome has become a powerful tool for metazoan phylogenetic and evolutionary analysis [21–24] because of its small size, the presence of high copy numbers, strict orthologous genes, rare recombination, and high evolutionary rate [22,25,26]. It has been applied in many insect groups to solve their phylogenetic relationships, including the whole Coleoptera [27–29] and some special groups (e.g., [30,31]), but not exclusively for Tenebrionoidea until now. With the newly sequenced mitochondrial genome and other available data in the public database (NCBI), we are going to investigate the phylogenetic relationship between the new species with others of Tenebrionoidea. Fortunately, the mitochondrial genome of *O. bipunctata* (Fabricius, 1775) (Melandryidae: Osphyinae) is available as a single sequence of the genus *Osphya* documented in NCBI, which makes it possible to test whether the new species can be grouped with the latter species into a monophyletic clade or not, thereby confirming the phylogenetic position of the new species.

China, covering a vast geographical area with diverse landscapes and located in a transitional zone between Palaearctic and Oriental regions, harbors giant biodiversity resources [32]. Although dozens of thousands of beetles have been recorded there, there is little doubt that some taxa remain to be discovered. As evidenced by a cursory look at most modern revisionary work, the number of "known" species is only a small subset of the number of "existing" species. This problem is aggravated by the biodiversity crisis and accelerating rates of biological extinction [33]. The global decline of insects and their abundance is becoming drastic due to climate change and human-derived perturbations [34], and it is probable that many species are vanishing without ever having been collected. Therefore, inventory work and targeted collecting (with specimens suitable for both morphological and molecular work) remain essential, and thus a priority, for the future of systematic studies. In this study, we discover and report a new species of *Osphya* based on both morphological and molecular evidence, which will make this widespread but species-poor genus better known, enrich the species diversity of beetles in the Chinese fauna in both generic and subfamilial levels, and will also increase interest in the other relevant research of enigmatic insects.

## 2. Materials and Methods

### 2.1. Materials

The studied material was collected from Shennongjia National Natural Reserve, Hubei Province, China. The type specimens (holotype and three paratypes) of the new species were deposited in the Museum of Hebei University, Baoding, China (MHBU). An additional specimen was fixed directly in 100% ethanol after collection and stored at −20 °C for molecular study.

### 2.2. Photography and Terminology

The specimens were dissected under the Nikon SMZ1500 microscope. The abdomen and aedeagus of the holotype were mounted on the same board as the body. The photography was processed by a Leica M205A stereomicroscope, and multiple layers were stacked using Combine ZM (Helicon Focus 5.3). The terminology in the species description follows Konvička [3,4]. The distribution information of the species was collected from the original publications and the material studied herein (Table S1). The distribution map was prepared by ArcGIS 10.6 and edited in Adobe Photoshop 2020.

### 2.3. DNA Extraction, PCR, and Sequencing

Total genomic DNAs were extracted using a DNeasy Blood & Tissue kit (QIAGEN, Beijing, China), according to the manufacturer's instructions. DNAs were stored at −20 °C for long-term storage and further molecular analyses and deposited at Hebei University (voucher number CAN0214).

Whole mitochondrial genome sequencing was performed using an Illumina Novaseq 6000 platform (Illumina, Alameda, CA, USA) with 150 bp paired-end reads at BerryGenomics, Beijing, China. The sequence reads were first filtered following Zhou et al. [35] and then assembled using IDBA-UD [36] under a similarity threshold of 98% and k values of a minimum of 40 and a maximum of 160 bp. The gene cox1 was amplified by polymerase chain reaction (PCR) using universal primers "5′-GGTCAACAAATCATA AAGATATTGG-3′ (LCO)" and "5′-TAAACTTCAGGGTGACCAAAAAATCA-3′ (HCO)" as "reference sequences" to acquire the best fit. The PCR cycling conditions comprised a predenaturation at 94 °C for 5 min and 35 cycles of denaturation at 94 °C for 50 s, annealing at 48 °C for 45 s, and elongation at 72 °C for 8 min at the end of all cycles. Geneious 2019.2 software [37] was used to manually map the clean readings on the obtained mitochondrial scaffolds to check the accuracy of the assembly. Gene annotation was performed by Geneious 2019.2 software [37] and MITOS Web Server (http://mitos.bioinf.uni-leipzig.de/index.py, accessed on 15 November 2021) [38]. The sequence generated in this study was deposited in GenBank with accession number (MW556727).

### 2.4. Phylogenetic Analyses

Mitochondrial genomes of 41 species from 16 different families of Tenebrionoidea were selected as ingroups, and those of 2 Cucujoidea species were chosen as outgroups (Table 1). All mitochondrial genomes (except the one sequenced in this study) were obtained from GenBank (accession numbers given in Table 1).

**Table 1.** Sequence information used in the phylogenetic analyses.

| Superfamily | Family | Species | GenBank Accession No. | References |
|---|---|---|---|---|
| Tenebrionoidea (ingroups) | Aderidae | Aderidae sp. | JX412763.1 | [28] |
| | Anthicidae | Anthicidae sp. | MH789722.1 | [27] |
| | | *Stricticollis tobias* (Marseul, 1879) | KX087350.1 | [29] |
| | Boridae | *Boros schneideri* (Panzer, 1795) | HQ232823.1 | [39] |
| | Ciidae | Ciidae sp. | JX412846.1 | [28] |
| | Melandryidae | *Mikadonius gracilis* Lewis, 1895 | JX412823.1 | [28] |
| | | *Osphya bipunctata* (Fabricius, 1775) | JX313675.1 | [28] |
| | | *Osphya sinensis* sp. n. | MW556727 | This study |
| | Meloidae | *Epicauta gorhami* (Marseul, 1873) | NC_036042.1 | [40] |
| | | *Hycleus marcipoli* Pan & Bologna, 2014 | NC_036044.1 | [40] |
| | | *Lytta caraganae* (Pallas, 1798) | NC_033339.1 | Unpublished |
| | | *Mylabris aulica* Menetries, 1832 | NC_036046.1 | [40] |
| | Mordellidae | *Mordella atrata* Melsheimer, 1845 | NC_013254.1 | [41] |
| | | *Mordellochroa milleri* (Emery, 1876) | KX087318.1 | [29] |
| | | *Tomoxia bucephala* (Costa, A., 1854) | KX087355.1 | [29] |
| | Mycetophagidae | *Mycetophagus quadripustulatus* (Linnaeus, 1761) | HQ232824.1 | [39] |
| | Mycteridae | *Hemipeplus* sp. | JX412852.1 | [28] |
| | | *Stilpnonotus mexicanus* (Thomson, 1860) | JX412811.1 | [28] |
| | Oedemeridae | *Ischnomera caerulea* (Linnaeus, 1758) | JX412790.1 | [28] |
| | | *Nacerdes carniolica* (Gistl, 1832) | KX087319.1 | [29] |
| | Prostomidae | *Prostomis* sp. | JX412787.1 | [28] |
| | Pyrochroidae | *Schizotus pectinicornis* (Linnaeus, 1758) | KX087342.1 | [29] |
| | Ripiphoridae | *Pelecotoma fennica* (Paykull, 1799) | NC_036277.1 | [29] |

Table 1. *Cont.*

| Superfamily | Family | Species | GenBank Accession No. | References |
|---|---|---|---|---|
| | Salpingidae | *Lissodema cursor* (Gyllenhal, 1813) | KX087307.1 | [29] |
| | | *Vincenzellus ruficollis* (Panzer, 1794) | NC_036274.1 | [29] |
| | Scraptiidae | *Anaspis* sp. | JX412856.1 | [28] |
| | | *Cyrtanaspis phalerata* (Germar, 1831) | KX087279.1 | [29] |
| | | *Scraptia* sp. | JX412825.2 | [28] |
| | Tenebrionidae | *Alphitobius diaperinus* (Panzer, 1797) | NC_049092.1 | [42] |
| | | *Asbolus verrucosus* LeConte, 1851 | NC_027256.1 | [43] |
| | | *Blaps rhynchoptera* Fairmaire, 1886 | NC_047449.1 | [44] |
| | | *Nalassus laevioctostriatus* (Goeze, 1777) | KT876905.1 | [45] |
| | | *Promethis valgipes* (Marseul, 1876) | MW201671.1 | [46] |
| | | *Scaphidema metallicum* (Fabricius, 1792) | KX087341.1 | [29] |
| | | *Strongylium suspicax* (Kolbe, 1894) | JX412780.1 | [28] |
| | | *Tenebrio molitor* Linnaeus, 1758 | NC_024633.1 | [47] |
| | | *Tribolium castaneum* (Herbst, 1797) | NC_003081.2 | [48] |
| | | *Ulomoides dermestoides* (Fairmaire, 1893) | NC_025332.1 | [49] |
| | | *Zophobas atratus* (Fabricius, 1775) | NC_041101.1 | [50] |
| | Tetratomidae | *Tetratoma fungorum* Fabricius, 1790 | NC_036276.1 | [29] |
| | Zopheridae | *Usechus lacerta* Motschulsky, 1854 | NC_036269.1 | [29] |
| Cucujoidea | Nitidulidae | *Carpophilus pilosellus* Motschulsky, 1858 | NC_046035.1 | [51] |
| (outgroups) | | *Omosita colon* (Linnaeus, 1758) | NC_050852.1 | Unpublished |

Data standardization and information extraction were performed by PhyloSuite v 1.2.2 [52]. The nucleotide sequences of the 13 protein-coding genes were analyzed. The protein-coding genes were aligned using the MAFFT v 7.313 plugins [53] and optimized using MACES in PhyloSuite v 1.2.2 [52]. Intergenic gaps and ambiguous sites were removed using Gblocks v 0.91b [54], then all protein-coding genes were concatenated in PhyloSuitev 1.2.2 [52]. The optimal partition schemes and the best-fit replacement models were selected by Model Finder [55], and the results are presented in Supplementary Tables S2 and S3. The "greedy" algorithm with branch lengths estimated as "linked" and the Bayesian information criterion were used.

Phylogenetic trees were constructed based on both maximum likelihood (ML) and Bayesian inference (BI). The ML phylogenies were inferred using IQ-TREE.v.1.6.8 [56] and BI phylogenies using MrBayes 3.2.6 [57], respectively. The ML phylogenetic analyses were performed with the ultrafast bootstrap (UFboot) algorithm with 1000 replicates. In BI phylogenetic analyses, $5 \times 10^6$ Markov chain Monte Carlo (MCMC) generations after reaching stationarity (average standard deviation of split frequencies <0.01) were used as the default settings, with estimated sample size >200 and potential scale reduction factor $\approx$ 1 [57]. Interactive Tree of Life (iTOL, http://itol.embl.de) was used to display, annotate, and manage the phylogenetic tree.

## 3. Results

### 3.1. Description of New Species

*Osphya sinensis* H. Liu et Y. Yang, sp. n.

Type material. Holotype: male (MHBU), China, Hubei Province, Shennongjia, Dajiuhu, Luoyanghe, 31°34′38″ N, 110°08′16″ E, 31.v.2019, leg. Ping Wang. Paratypes: China, Hubei Province, Shennongjia: 1 male, 1 female (MHBU), Muyu, Yanjiawan, 1661 m, 31°28′38″ N, 110°26′03″ E, 30.vi.2022, leg. Junbo Tong and Xueying Ge; 1 female (MHBU), Muyu, Longjiangping, 110°23′25″ E, 31°30′07″ N,4.vi.2018, leg. Ping Wang.

Additional material. China, Hubei Province, Shennongjia, Dajiuhu, Dongxi, 31°32′22″ N, 110°07′19″ E, 1.vii.2019, leg. Ping Wang (MHBU, CAN0214).

Diagnosis. *Osphya sinensis* sp. n. (Figures 1–3) is characterized by the following combination of character states: (1) large-sized body; (2) metallic blue body coloration,

except antennomeres I–III and VIII–XI orange, pronotum uniformly orange in female, or with a heart-form dark blue marking in the center of the disc in males, legs bicolored, mixed blue with orange, abdomen blue in males while orange in females; (3) all tibial spurs serrulate laterally; (4) aedeagus with parameres weakly narrowing from base to apex, dorsally emarginate on apical third part; (5) abdominal ventrite V shallowly and triangularly emarginate in middle of posterior margin and tergite V tapered apically in females, while both widely rounded at apices in males.

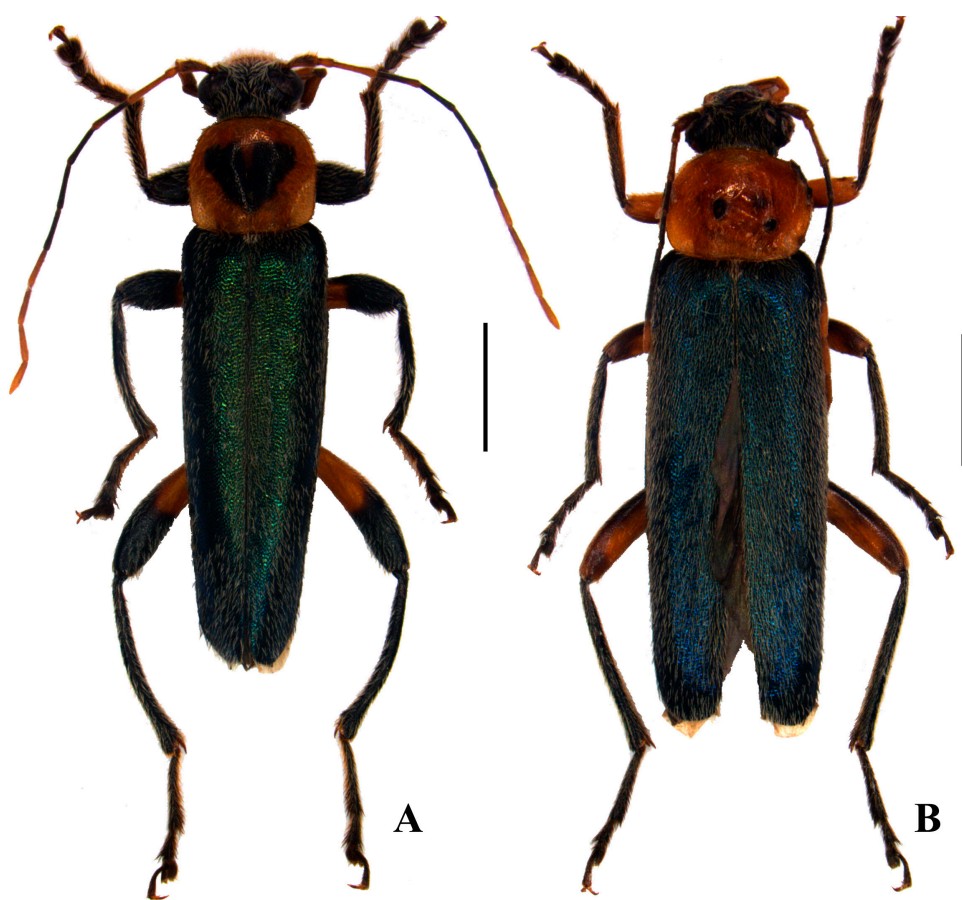

**Figure 1.** Habitus of Osphya sinensis sp. n., dorsal view: (**A**) male; (**B**) female. Scale bar: 2.0 mm.

Etymology. The specific name is derived from Latin *Sinensis* (Chinese), referring to its type locality in China.

Comparisons with congeners. This species is most similar to *O. superba* Pic, 1927 (located in Chapa, Vietnam), but differs from the latter in the following characteristics: the body is smaller (9.5–10.0 mm in length), the pronotum has a large heart-form dark blue marking in males or uniformly orange in females, scutellum is dark blue, and legs are bicolored and mixed blue with orange. Unlike in the new species, O. superba has a larger body (12.0 mm in length), the pronotum bears black markings in the middle line and on both sides of the disc, and the scutellum and legs are uniformly orange.

Description. Male (Figure 1A). Body length: 10.0 mm, width: 2.0 mm. Body dark blue and moderately shining, mouthparts orange, black at apices of mandibles, more or less darkened at maxillary and labial palpomeres, antennae orange at antennomeres I–III and VIII-XI, which slightly darkened dorsally, pronotum orange, darkened at the middle of anterior and posterior margins, with a heart-form dark blue marking in the center of the disc, legs orange at coxae, basal halves of femora and ventral sides of protibiae, elytra green blue and strongly shining, and body densely covered with gray pubescence, except for orange on the pronotum.

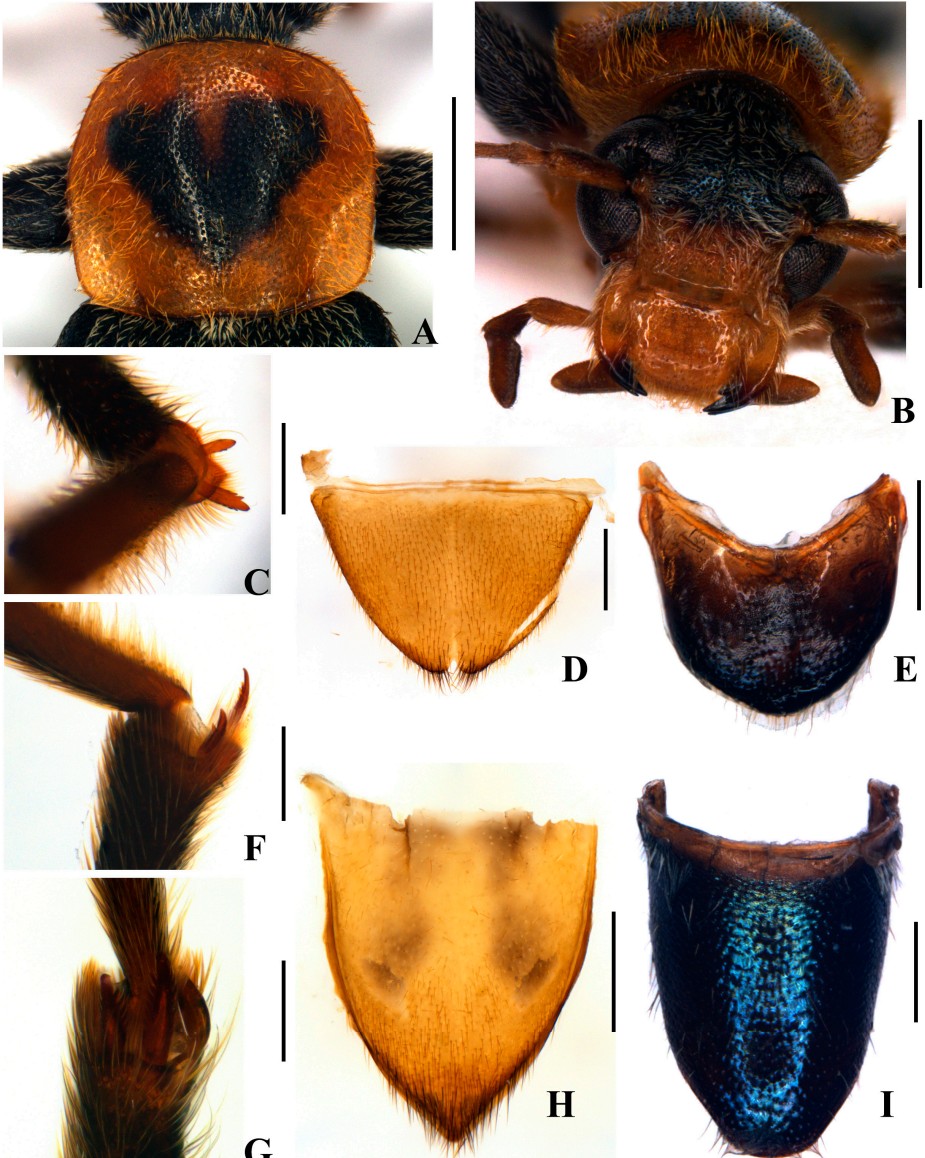

**Figure 2.** Osphya sinensis sp. n. (**A–C,E–G,I**) male; (**D,H**) female: (**A**) pronotum, dorsal view; (**B**) head, frontal view; (**C**) protibial spurs, dorsal view; (**D,E**) abdominal ventrite V, ventral view; (**F**) mesotibial spurs, lateral view; (**G**) metatibial spurs, dorsal view; (**H,I**) abdominal tergite V, dorsal view. Scale bars (**A,B**) 1.0 mm; (**D–E,H,I**) 0.5 mm; (**C,F,G**) 0.2 mm.

Head small, abruptly constricted behind eyes, obviously narrower than pronotum, surface finely and densely punctate; mandibles (Figure 2B) bidentate near apices; eyes large, emarginate in middle of the inner margin; terminal maxillary palpomere cultriform, about 3.0 times as long as wide, sharp at inner margins; antennae extending over elytral mid-length, antennomeres II short, about 1.5 times as long as wide at apices, III about 3.0 times longer than II, IV, and V nearly as long as III, VI longest, VII–X decreasing in length, XI slightly longer than X and pointed at apices.

Pronotum (Figure 2A) transverse, about 1.15 times wider than long, widest nearly in middle, anterior, and lateral margins arcuate, posterior margin nearly straight, anterior angles confluent with anterior and lateral margins, posterior angles subrounded, middle longitudinal line weakly indicated in the middle part, disc strongly convex in the center of the disc, surface punctate as that on the head, latero-basal impressions shallow and moderately traceable. Scutellum rounded at the apex.

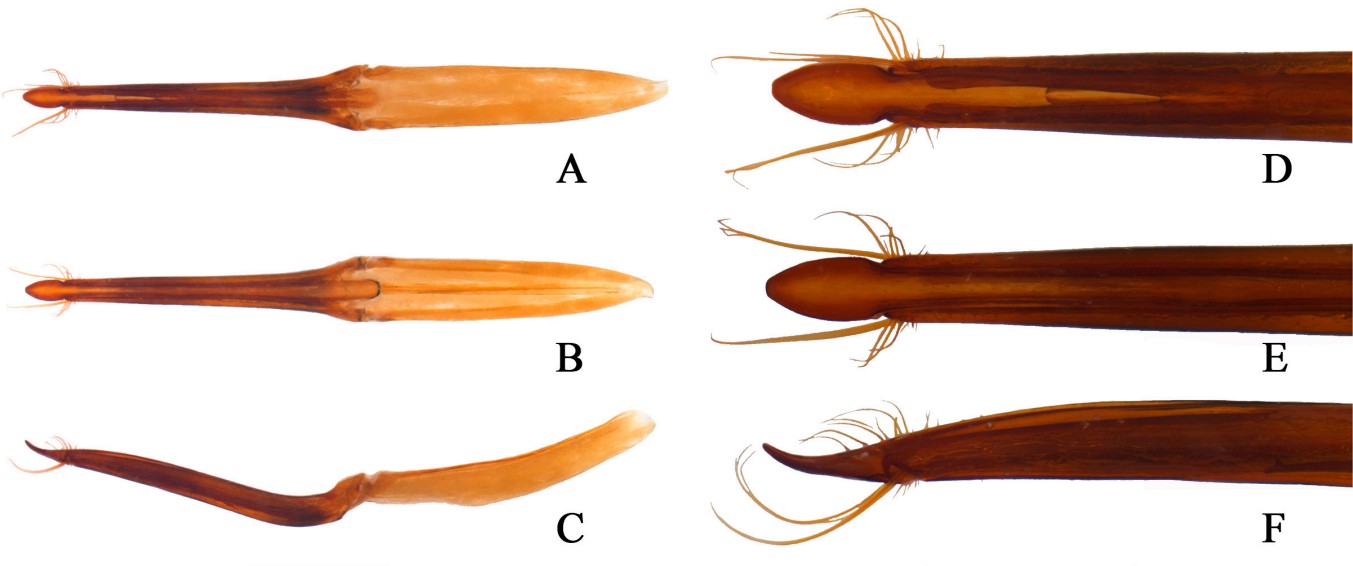

**Figure 3.** *Osphya sinensis* sp. n. (**A–C**) the whole of the aedeagus; (**D–F**) the apical half part of the aedeagus. (**A**,**D**) dorsal view; (**B**,**E**) ventral view; (**C**,**F**) lateral view. Scale bars (**A–C**) 1.0 mm; (**D–F**) 0.5 mm.

Elytra is about 3.3 times longer than the pronotum, about 3.0 times as long as the humeral width, with lateral margins obliquely converging posteriorly, apices rounded, and surface more coarsely and densely punctate than that on the pronotum.

Leg slender, femora moderately inflated, tibiae slightly curved inwards at apical parts, the relative ratio of pro-, meso-, and meta-tibiae in length as 25:32:38, each tibia with a pair of subequal spurs at apices, the spurs serrulate laterally (Figure 2C,F,G), meta-tarsomeres I longer than the combined length of others, all claws bidentate.

Abdominal ventrite V (Figure 2E) transverse, tergite V (Figure 2I) elongate, both rounded at posterior margins. Aedeagus (Figure 3) slender, about 0.6 times as long as body length, basal piece nearly as long as parameres, both weakly narrowing from base to apex; parameres completely separated ventrally, conjoint dorsally and emarginate at apical third part, covered with a tuft of long, curved and yellow setae on each side of the apex; median lobe nearly as long as parameres, articulate at the point where basal piece and parameres meet, with apex lanciform and extruding over parameres.

Female (Figure 1B). Body more robust and broader than males. Coloration is similar to that of males, but the pronotum, meso- and metasterna, and abdomen are uniformly orange, femora mostly orange and only darkened along the dorsal sides. Antennae shorter, extending to the basal third length of elytra. Pronotum wider, 1.35 times as wide as long. Tibiae straight and femora normal, not inflated. Abdominal ventrite V (Figure 2D) shallowly and triangularly emarginated in the middle of the posterior margin, tergite V (Figure 2H) tapered apically.

Variation within type series. Body length 9.5–10 mm, width 1.9–2.5 mm.

Distribution. China, Hubei Province, Shennongjia National Natural Reserve.

Remarks. The additional material used for the molecular study was seriously damaged in the experiment, so it was not designated as the type specimen.

### 3.2. Phylogenetic Analyses

In both BI and ML analyses (Figures 4 and S1), the monophyly of Tenebrionoidea is well supported with high supporting values (BS = 100, PP = 1). Additionally, the monophylies of the related families, including Tenebrionidae, Meloidae, Oedemeridae, Anthicidae, Scraptiidae, Salpingidae, and Mordellidae, which with two or more analyzed species are all highly supported (BS = 100, PP = 1). This suggests that the phylogenetic relationships of Tenebrionoidea are generally reconstructed well enough based on the

mitochondrial genome data, which can be relied on to investigate the placement of the new species.

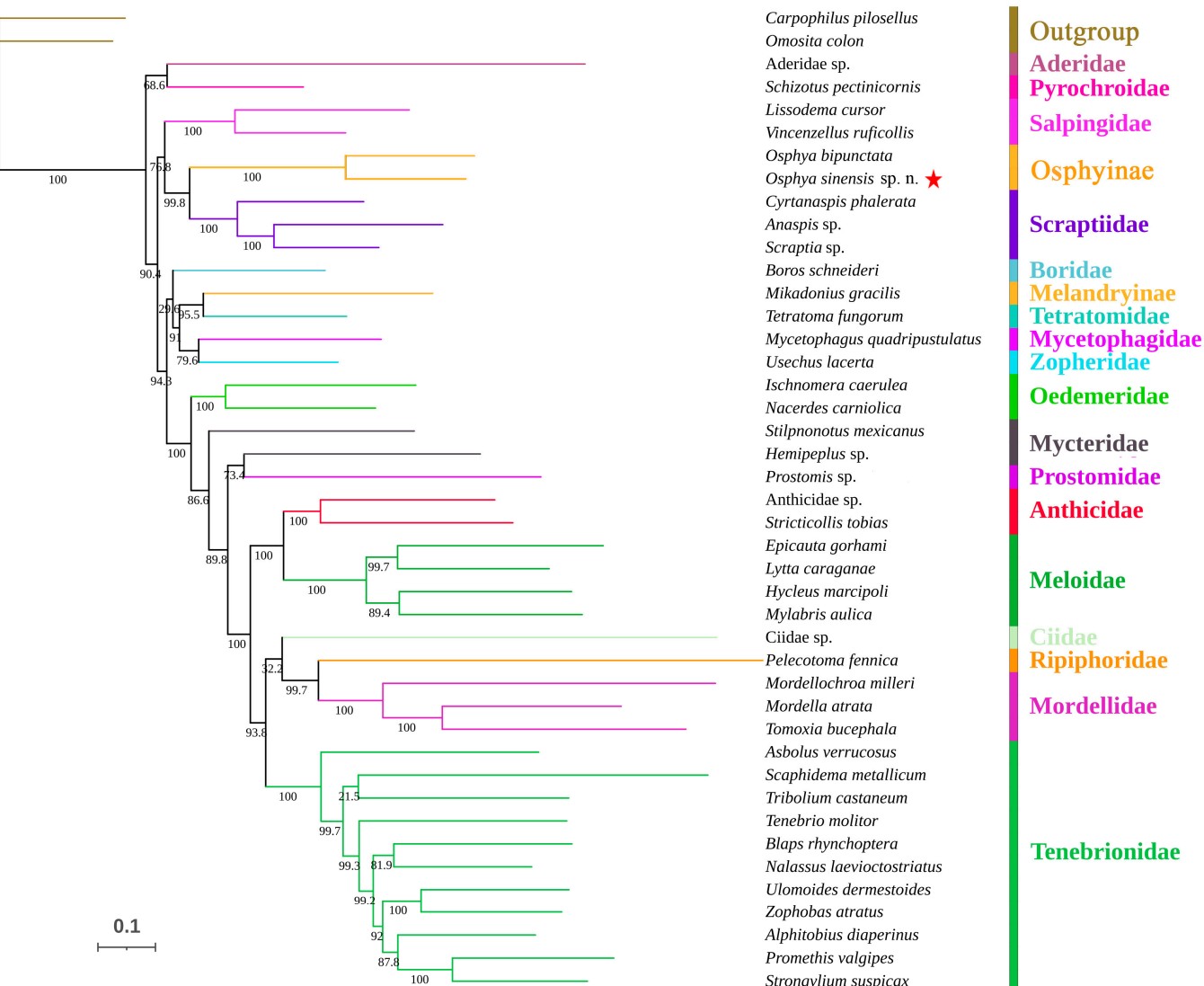

**Figure 4.** Phylogenetic tree of Tenebrionoidea produced from the ML analysis based on 13 PCGs (BS values indicated in each clade). The red star denotes *Osphya sinensis* sp. n.

*Osphya sinensis* sp. n. is consistently grouped with *O. bipunctata* (Fabricius, 1775) into a monophyletic clade, which is strongly supported (BS = 100, PP = 1). These two representative species of Osphyinae were recovered sister to Scraptiidae with high supporting values (BS = 100, PP = 1), instead of Melandryinae (represented by *M. gracilis*), which is either grouped with Tetratomidae (*T. fungorum*) by BI analysis (Figure 4) or in an unclear relationship with other lower Tenebrionoidea by ML analysis (Figure S1). This indicates that Melandryidae, with two component subfamilies (Osphyinae and Melandryinae) separate in different clades, is a paraphyly based on the mitochondrial genome-based phylogeny.

## 4. Discussion

### 4.1. Separate Status of the New Species

Prior to this study, a total of 12 *Osphya* species were hitherto known and distributed in the Oriental and eastern and southern Palaearctic regions [1,3] (Figure 5 and Table S1). Among these species, five species are distributed in N India (*O. albofasciata* Champion [7], *O. harmandi* Pic [13], *O. dissimilis* Champion [9], *O. nigriventris* Champion [8], and *O. nigroapicalis*

Pic [11]), two in Taiwan Island (*O. trilineata* Pic [10] and *O. formosana* Pic [14]) and N Vietnam (*O. rufa* Pic [15] and *O. superba* Pic [15]), one in Japan (*O. orientalis* (Lewis) [5]), S Myanmar (*O. melina* Champion [7]) and S India (*O. nilgrica* Champion [7]), respectively.

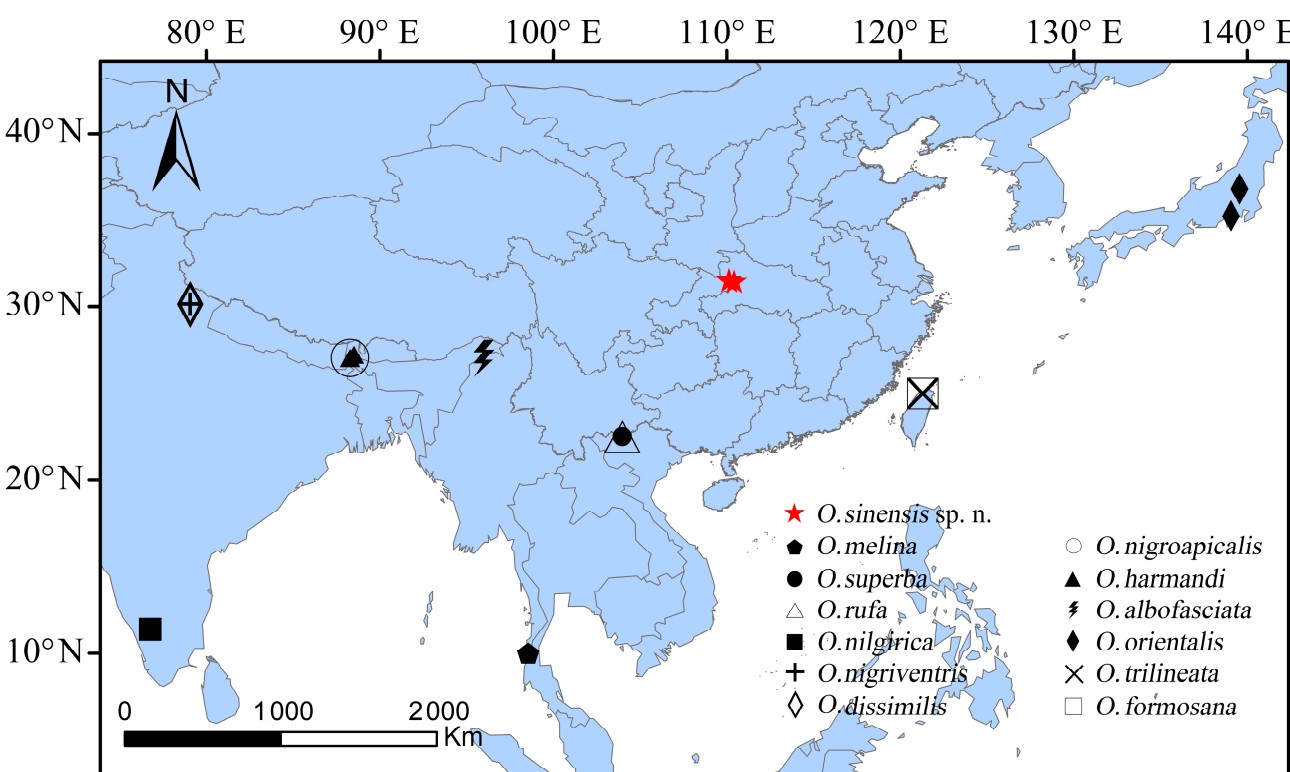

**Figure 5.** Distribution of the *Osphya* species in the Oriental and eastern and southern Palaearctic regions.

In contrast with the European species [3,4,17], these species are quite poorly known. According to their original descriptions [5,7–11,13–15], the new species, *O. sinensis* sp. n., could be easily differentiated from the above species by the following key.

**Key to the *Osphya* species in the Oriental and eastern and southern Palaearctic regions**
1. Body large-sized, more than 10 mm; elytra uniformly green or blue, with strongly metallic shine . . . . . . . . . . . . . . . . . . . . . . . . . . . . . . . . . . . . . . . . . . . . . . . . . . . . . . . . ..2
- Body small or middle-sized, less than 8 mm; elytra black, yellow or reddish brown, or bicolored, at most with weakly metallic shine . . . . . . . . . . . . . . . . . . . . . . . . . . . . . . . . ..3
2. Body 12 mm in length; body green, scutellum and legs uniformly orange . . . . . . . . . . . . . . . . . . . . . . . . . . . . . . . . . . . . . . . . . . . . . . . . . . . . . . ..*O. superba*
- Body 9.5–10 mm in length; body blue, scutellum dark blue, legs bicolored, mixed blue with orange.. . . . . . . . . . . . . . . . . . . . . . . . . . . . . . . . . . . . . . . . *O. sinensis* sp. n.
3. Body covered with reddish-brown pubescences . . . . . . . . . . . . . . . . . . . . . . . . . . . . . .4
- Body covered with gray or black pubescences . . . . . . . . . . . . . . . . . . . . . . . . . . . . . ...5
4. Body 7 mm in length; elytra uniformly reddish brown . . . . . . . . . . . . . . . . . . . . . . . . *O. rufa*
- Body 8 mm in length; elytra reddish brown, black at apices . . . . . . . . . . . . . . . *O. nigroapicalis*
5. Elytra decorated with gray longitudinal or transverse bands . . . . . . . . . . . . . . . . . . . . 6
- Elytra uniformly colored, without any longitudinal or transverse band . . . . . . . . . . . . . . ...8
6. Body 6–6.5 mm in length; each elytron with a longitudinal band along the suture and two transverse bands on the disc... . . . . . . . . . . . . . . . . . . . . . . . . . . . . . . . . . . . . . .*O. orientalis*
- Elytra with gray pubescent bands along sutures and on each side of the middle disc . . . . . . . . 7
7. Body slender, 6 mm in length; antennomeres IV and V relatively long . . . . . . . . . ...*O. trilineata*
- Body stouter, 5 mm in length; antennomeres IV and V relatively short . . . . . . . . . ..*O. albofaciata*
8. Body mostly or uniformly yellow . . . . . . . . . . . . . . . . . . . . . . . . . . . . . . . . . . . . . . . ...9

- Body black . . . . . . . . . . . . . . . . . . . . . . . . . . . . . . . . . . . . . . . . . . . . . . . . . . . . . . . . . . . . . .10
9. Body small and 4.25 mm in length, uniformly yellow . . . . . . . . . . . . . . . . . . . . . . . *O. melina*
- Body middle-sized and 7.5 mm in length, mostly yellow, black at anterior and middle tibiae,
apex of terminal maxillary palpomere, and antennomeres IV-XI . . . . . . . . . . . . . ...*O. nigriventris*
10. Body 5.5 mm in length; antennomeres II, IV, and V subequal in length, III, VI, and VII slightly
longer, VIII–XI stouter . . . . . . . . . . . . . . . . . . . . . . . . . . . . . . . . . . . . . . . ...*O. dissimilis*
- Body longer than 6.0 mm; antennae unlike above . . . . . . . . . . . . . . . . . . . . . . . . . . . 11
11. Body 7.0 mm in length; elytra with weakly metallic shine . . . . . . . . . . . . . . ...*O. formosana*
- Elytra without metallic shine . . . . . . . . . . . . . . . . . . . . . . . . . . . . . . . . . . . . . . . .12
12. Body slender and 8.0 mm in length; head smaller . . . . . . . . . . . . . . . . . . . . *O. harmandi*
- Body stouter and 6.0 mm in length; head broader . . . . . . . . . . . . . . . . . . . . . *O. nilgirica*

*4.2. Phylogenetic Position of the New Species*

The new species, *O. sinensis* sp. n., matching the diagnosis of *Osphya* in morphology well, is consistently grouped with *O. bipunctata* to form a monophyletic clade based on the mitophylogeny of Tenebrionoidea, which is solidly and highly supported. Although *Osphya* is discovered in the Chinese mainland for the first time, its occurrence is in accordance with the geographical distribution of this genus, which partly occurs in the Palaearctic and Oriental regions [20] covering the Chinese fauna. What is noted, the locality of the new species in mainland China fills the geographical gap of *Osphya* between the western and easternmost parts of the Palaearctic Region. As a widely distributed genus in the Northern Hemisphere [20], it would be more interesting to investigate the formation of spatial distribution patterns of *Osphya*, which are highly relevant to the contemporary environment and historical factors, but this is not within the scope of this study.

Further, the phylogenetic analysis of Tenebrionoidea of mitogenomic data showed that the monophyly of Melandryidae is not recovered, which is in agreement with some specialists (e.g., [19]), is also congruent with the results of other molecular phylogenetic studies [28,58], but is discordant with the opinions of the modern coleopterists [1,2,20,59,60]. This probably resulted from a lack of a satisfactory definition for the family, which allows many non-melandryids to be placed in this group [59–62]. Crowson [59] stated that both the imaginal and larval characterizations of this family were unsatisfactory, so there had been historically wide disagreements over its constitution. In the classification of Pollock [19], only the two subfamilies, Melandryinae and Osphyinae, were recognized, which is a much more restricted concept of the family Melandryidae than that presented by Csiki [61], Crowson [59], Arnett [62], and Lawrence and Newton [60]. This classification was adopted in the latest catalog of Palearctic Coleoptera by Nikitsky and Pollock [20]. The Osphyinae can be distinguished from Melandryinae in the characteristics of both adults and larvae [1], which can be considered as synapomorphies of each lineage.

However, Osphyinae is recovered sister to Scraptiidae, congruent with the results of other molecular phylogenetic results, such as Hunt et al. [58], Gunter et al. [63], and Timmermans et al. [28]. Additionally, Melandryinae seems more related to Tetratomidae, which is consistent with the result of Hunt et al. [58]. Crowson [59] suggested that Melandryidae be close to Tetratomidae due to some affinities in adult structures also shared with Scraptiidae, Mordellidae, and Ripiphoridae, but their similarities were argued to be convergent by Lawrence [64]. Thus, a more comprehensive systematic study of Melandryidae is needed to verify their relationships to other lower Tenebrionoidea, suggested by Pollock [19].

## 5. Conclusions

The genus *Osphya* is discovered in the Chinese mainland for the first time, with a new species, *O. sinensis* sp. n., reported from Shennongjia National Natural Reserve, Hubei Province. Among the *Osphya* species that are distributed in China and adjacent countries or areas, the new species is most similar to *O. superba* located in N Vietnam but definitely differs from the latter species in the body size and coloration. Except for the morphological evidence, the placement of the new species in the genus *Osphya* is further confirmed by the mitophylogenetic analyses of Tenebrionoidea, and a sister relationship is recovered between

the new species and *O. bipunctata*. However, its phylogenetic position in Melandryidae remains uncertain due to the paraphylum of the family, with the two component subfamilies Osphyinae and Melandryinae separate in different clades. Nevertheless, the new species is definitely attributed to *Osphya* of Osphyinae, but larger scale studies with more locus and taxon sampling are still needed to reconstruct more comprehensive phylogenies in order to achieve a better resolution of the relationships of Osphyinae to Melandryinae and other families of Tenebrionoidea.

**Supplementary Materials:** The following supporting information can be downloaded at: https://www.mdpi.com/article/10.3390/d15020282/s1, Figure S1: Phylogenetic tree of Tenebrionoidea inferred from the BI analysis based on 13 PCGs (PP values indicated in each clade); Table S1: Distribution information of the *Osphya* species in the Oriental and eastern and southern Palaearctic regions. Table S2: The best partitioning schemes and models for the maximum likelihood (ML) method; Table S3: The best partitioning schemes and models for the Bayesian inference (BI) method.

**Author Contributions:** Conceptualization, H.L. and Y.Y.; methodology, L.Y. and J.T.; software, L.Y.; validation, H.L., Z.P. and Y.Y.; formal analysis, L.Y. and Y.Y.; investigation, P.W., J.T. and G.W.; data curation, L.Y. and Y.Y.; writing—original draft preparation, L.Y. and Y.Y.; writing—review and editing, L.Y. and Y.Y.; visualization, L.Y. and J.T.; supervision, H.L. and Y.Y.; project administration, H.L. and Y.Y.; funding acquisition, H.L., G.W. and Y.Y. All authors have read and agreed to the published version of the manuscript.

**Funding:** This work was supported by the National Natural Science Foundation of China (No. 32270491), the Natural Science Foundation of Hebei Province (No. C2022201005), the Excellent Youth Scientific Research and Innovation Team of Hebei University (No. 605020521005) to Y.Y., the Biodiversity Survey, Observation and Evaluation Project of Hubei Province to G. W., and the Interdisciplinary Research Program of Natural Science of Hebei University (No. DXK202103) to H.L.

**Institutional Review Board Statement:** Not applicable.

**Data Availability Statement:** The sequence generated in this study is available in GenBank with accession number (MW556727).

**Acknowledgments:** We are grateful to Xueying Ge for her help in the DNA extraction, to Tong Liu for the preparation of the distribution map, and to Fang Zhang for the preparation of the specimens.

**Conflicts of Interest:** The authors declare no conflict of interest.

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
