# Peer review of "First Record of Osphya (Melandryidae: Osphyinae) from Chinese Mainland Based on Morphological Evidence and Mitochondrial Genome-Based Phylogeny of Tenebrionoidea"

_diversity, doi:10.3390/d15020282_

Round 1

Reviewer 1 Report

The submitted MS dedicated to description of a new species of the genus Osphya and mitochondrion based phylogeny of Osphya and related taxa. In general, them MS is well-written, however it consist a number of minor errors and some general moments to be corrected prior acceptance. See below.

Line 43.

“China except for two species from Taiwan island“

Please, specify these two species here.

Lines 50-51

“Further comparison with others from the adjacent areas showed it as an unknown species of Osphya“

I would recommend to specify here names of the “other” species

Lines 85-87

“The gene cox1 was amplified by polymerase chain reaction (PCR) using universal primers as “reference sequences” to acquire the best-fit.

This part is not clear for me. Do you additionally amplified COI gene and sequenced it using traditional Sanger sequencing as reference? If so, please add the corresponding information in Materials and Methods section (PCR conditions and temperature profile, primers, how and where sequencing was performed, etc,)

Line 88

readings onthe obtained

Add space

Line 125

“vi.30.2022”

Should be 30.vi.2022

Line 201

elytra.Pronotum

Add space

Lines 215-216

“This indicates that Melandryidae is a paraphyly but monophyly based on the mitochondrial“…

Absolutely not clear, what do you mean saying Melandryidae is a paraphyly but monophyly?  

Lines 226-228

“…O. nigroapicalis Pic, 1921a [11]), two in Taiwan Island (O. trilineata Pic, 1910 [10]and O. formosana Pic, 1927a [14]) and N. Vietnam (O. rufa Pic, 1927b [15]and O. superbaPic, 1927b

228 [15])…”

Please note that author and year of the description after Latin names is NOT a reference. Thus, it can ‘not be Pic, 1927a, 1927b, etc., only a year Pic, 1927;

Add space after [10];

Add space after [15];

Add space after O.superba (O. superbaPic)

Lines 272-273

“In terms of this geographical distribution pattern, it is most probable for the new species to be placed in Osphya…”

This sentence in the discussion section somewhat confusing. The placement of the new species within Osphya genus was confirmed by morphology and molecular phylogeny. Thus, why do you use “it is most probable for the new species to be placed in Osphya”? I would recommend rephrasing this sentence. E.g. to note that occurrence of Osphya in China are in accordance with geographical distribution of the Osphya genus members.  

Author Response

Please see the details in the response letter.

Reviewer 2 Report

The manuscript under review is devoted to the application of an integrative approach to the study of representatives of the genus Osphya of the family Melandryridae.

The manuscript is built according to the classical type for works of this type. Morphological features have been studied, a key has been compiled to identify representatives of the genus in the study area.

Molecular data also confirm the validity of the description of a species new - Osphya sinensis H. Liu et Y. Yang, sp. n.

In general, there are no significant comments on the manuscript. The work may be published in the Diversity.

Author Response

please see the details in the reponse letter.

Reviewer 3 Report

The manuscript is prepared very professionally, the introduction and discussion are supported by a large number of references, the methodology and results are presented in a high-quality manner. However, I have a few comments and questions.

L86: Could you be more specific about the universal primers for COXI? Still, there are several variants.

L88: correct „onthe“

L195: correct „Scale barsA“

L201: correct „elytra.Pronotum“

L224-L230: I think it would be better to write North or N without a period than “N.”

L240: Body 10 mm in length – I always prefer that the keys show the full range of sizes that the species can reach. The range would be able to estimate - certainly there is at least a small variability in lengths between the holotype and the paratypes - so the mean and standard deviation can be calculated. If we simply assume a normal distribution, then the range of values that the body size would reach for 95% of individuals (after subtracting 5% of extreme values) at a given mean and a given standard deviation could be given.

L280: correct „phyolgeneitc“

L281: correct „coleoptersits“

Author Response

(The authors gave the same response as above.)

Reviewer 4 Report

Dear Authors, this manuscript contains very interesting and important information. However there are some point that need to be addressed. Please find my comments into the attached file.

Author Response

I made a point-by-point response to each comment, please see the details in the pdf.

Reviewer 5 Report

The article contributes to the study of Melandryidae. The taxonomic part corresponds to the Code of Zoological Nomenclature. Description is correct. Genetic analysis performed correctly.

My comments:

1. Keywords should not repeat the words from the Title.

2. It is necessary to give a map of the distribution of species of the genus Osphya in East Asia.

3. It is necessary to justify the assignment of a new species to the genus Osphya.

Author Response

Please see the details in the attached file. 

Reviewer 6 Report

The topic of the MS “First record of Osphya (MelandryridaeMelandryidae: Osphyinae) from Chinese mainland based on morphological evidence and mitochondrial genome-based phylogeny of Tenebrionoidea” is relevant to the journal “MDPI Diversity”. The paper is original (not published as far I know) and provide new data concerning Melandryidae. The paper adds to science one new (and first) Osphya species from China. The paper is useful and well-imaged contribution in the subject area. The conclusions in the paper and placement of the specimen into the genus are ok and justified. The confirmation of the phylogenetic position with use of mitochondrial genome was not necessary for description of new species, but such analysis is probably interesting for itself and fundamental science in general. The genetic comparison would be more useful, if authors compare Osphya sinensis with representatives of the genus from Eastern Asia (Japan, Taiwane etc.) and could determine the relations among these species.

The paper is scientifically correct.  References are appropriate. The existing list of references is ok, and I don’t see any excess citations.

My possible recommendation for the MS is following:

1. split the chapter “4.2. Phylogenetic position of the new species” into two sections, e.g. “4.2. Phylogenetic position of the new species” and “4.3. Some data on phylogenetic analysis of Tenebrionoidea”

So, in sum, the MS is clear and useful, it is good and can be published.

Author Response

Please see the details in the attachment. 

Round 2

Reviewer 4 Report

Dear authors,

Please fix the typo error at 2.1 Materials (studied)

Also, at 3.2. paragraph, since the species Mikadonius gracilis and Tetrotoma fungorum are already mentioned in the text please change with M. gracilis and T. fungorum

At the references section, changes are still needed. For example, after the year (in bold), the volume should be in italics. Also the years should not be followed by letters, e.g., 2017a, 2017b.

Concerning the years after the latin names of species (e.g., O. formosana Pic, 1927) it is rather rare to mention it in Entomology papers. If you want to emphasize the years that they have found the species please add a sentence that will approximately contain the years and not after their latin name. In the case of the Table you can add an extra column that will contain the year.

Best regards

Author Response

Thanks for your comments. Please see the response in the attachment. 
